# The Isotopx NGX and the ATONA Faraday Amplifiers

Stephen E Cox[1], Sidney R Hemming[1, 2], and Damian Tootell[3]

[1]Lamont-Doherty Earth Observatory, the Earth Institute at Columbia University, 61 Route 9W, Palisades, NY 10964, USA
[2]Department of Earth and Environmental Sciences, Columbia University, New York, NY 10027, USA
[3]Isotopx, Ltd., Dalton House, Dalton Way, Middlewich CW10 0HU, UK

**Correspondence:** Stephen E Cox (stephen@stephencoxgeology.com))

**Abstract.** We installed the new Isotopx ATONA Faraday cup detector amplifiers on an Isotopx NGX mass spectrometer at Lamont-Doherty Earth Observatory in early 2018. The ATONA is a capacitive transimpedance amplifier, which differs from the traditional resistive transimpedance amplifier used on most Faraday detectors for mass spectrometry. Instead of a high gain resistor, a capacitor is used to accumulate and measure charge. The advantages of this architecture are a very low noise floor, rapid response time, stable baselines, and very high dynamic range. We show baseline noise measurements and measurements of argon from air and cocktail gas standards to demonstrate the capabilities of these amplifiers. The ATONA exhibits a noise floor better than a traditional $10^{13}$ $\Omega$ amplifier in normal noble gas mass spectrometer usage, superior gain and baseline stability, and an unrivaled dynamic range that makes it practical to measure beams ranging in size from below $10^{-16}$ A to above $10^{-9}$ A using a single amplifier.

## 1 Introduction

The design of analog ion collectors for mass spectrometry has changed strikingly little for seventy years. Early instruments already employed much of the detector technology we recognize today, including multiple collectors, secondary electron suppressors, and electronic circuits that employed high-value resistors (resistor transimpedance amplifiers, or RTIA) to amplify small currents to measurable voltages (e.g., Nier, 1940, 1947). Between the 1950s and 1980s, as the field of isotope geochemistry shifted from home-brewed instruments to commercial ones, available noble gas mass spectrometers consolidated around a design based on the Reynolds mass spectrometer using a "Nier-type" ion source, a fixed accelerating voltage, a variable magnetic field, and a single pair of collectors consisting of an analog electron multiplier (later an ion counting multiplier) and a Faraday cup, intended to be used separately for signals of different sizes (e.g., Reynolds, 1956; Bayer et al., 1989; Renne et al., 1998; Burnard and Farley, 2000). Since around 2010, multicollection has come back into vogue as improvements in electronic noise and stability have mitigated the problems of comparing beams measured on two separate amplifiers, and the field has sought ways to minimize the uncertainty conferred by the fitting of gas evolution trends in order to calculate isotopes ratios at the time of sample inlet (e.g., Mark et al., 2009; Coble et al., 2011).

The shift toward multicollection has been accompanied by a diversification of the collector technologies available, with new ion counting multipliers built with a geometry that allows multicollector spacing, and new RTIA Faraday amplifiers employ-

ing higher-value resistors in order to take advantage of the $\sqrt{R}$ relationship between normalized signal noise and resistance (e.g., Zhang et al., 2016). These advances are not without trade-offs, however. For one, multicollection requires the use of wider flight tubes and larger collector blocks that increase the volume of static vacuum instruments, reducing their effective sensitivity; some applications may still benefit from the use of small volume single collector instruments, for which fast, high-dynamic-range detectors are particularly valuable. One some multicollector instruments, the use of ion counting multipliers in the detector position for large beams ($^{40}$Ar, for example) allows the measurement of very small samples but limits the dynamic range (Jicha et al., 2016). Instruments using high-value resistor amplifiers to achieve the same goal also suffer from a loss of dynamic range, although it is not as severe, but additionally suffer from long settling times (large Tau), baseline instability, and drift in gain calibration. The settling time problem is less severe on multicollectors that do not need to peak hop, but can still affect signal stability during the start of a measurement. These problems have limited the use of such collectors for decades, but the cost-benefit calculation has shifted due to improving electronic stability and new techniques for dealing with the Tau-correction (Zhang et al., 2016), as well as a cultural shift in the priorities of noble gas geochemistry labs toward young, small samples and higher precision (e.g., Wijbrans et al., 2011; Jicha et al., 2012; Mark et al., 2017; Rose and Koppers, 2019).

However, the desire to measure young samples well has not displaced the need to measure old samples very precisely (Sprain et al., 2015), to measure large amounts of noble gas in ice core and water samples (Lu et al., 2014), and to measure extreme abundance ratios such as those that are typical in $^{3}$He/$^{4}$He analyses (Espanon et al., 2014). The ideal collector, therefore, has not just a low noise floor and high sensitivity, but also a high dynamic range, the ability to switch between low and high signals with no memory, and a stable, precisely measurable gain bias between each detector. Resistor Faraday amplifiers have a fairly restricted dynamic range, with the ability to reliably measure signals over only about five orders of magnitude. Ion counting multipliers are only able to measure small signals, and suffer from significant nonlinearity at the upper and lower ends of their useful range. Analog multipliers have a much higher dynamic range, about eight orders of magnitude, but suffer from both nonlinearity and relatively short timescale gain drift. In addition, electron multipliers wear quickly and are both expensive and vulnerable to damage from vacuum accidents and large ion beams. Faraday cups are extremely linear, quiet, resilient, and cheap to manufacture, so a technological solution that extends their useful dynamic range and sensitivity to small signals is highly desirable.

Mass spectrometers have always relied on transimpedance amplifiers, which consist of an active circuit element (usually an op-amp) that converts a small input current to a high output voltage (Figure 1). The capacitive transimpedance amplifier was developed decades ago, and was an option on such venerable devices as the Keithley 6512 Electrometer, which provided the option of feedback resistors or capacitors for current measurements. The advantage of the latter was seen as the high dynamic range, while the disadvantages were the accuracy and linearity. More recent work has demonstrated the promise of low noise and stability using feedback capacitors, but with serious limitations on dynamic range, linearity, and flexibility caused by the measurement of accumulated charge and the need to handle routine discharging of the feedback capacitor (Esat, 1995; Ireland et al., 2014). The new ATONA capacitive transimpedance amplifier developed by Isotopx maintains the high dynamic range

(effectively unlimited for noble gas measurements) and rapid response time of the earlier feedback capacitor devices while also delivering the linearity and accuracy more traditionally associated with resistor transimpedance amplifiers. The ATONA uses a proprietary extremely low leakage dielectric for the feedback capacitor combined with a cooled amplifier housing to reduce the leakage current, and consequent nonlinearity, to below 1 ppm. Unlike previous charge-mode amplifiers, the ATONA measures rate of change of the transimpedance amplifier output voltage and therefore the rate of change of the accumulated charge. The advantages of this setup, which can accurately measure extremely low signals without sacrificing stability or the ability to measure large signals, are significant for noble gas mass spectrometry and for mass spectrometry in general.

Noble gas mass spectrometers must measure an evolving signal due to the action of the instrument itself on the sample (Figure 2). Sample abundances are typically so small that the entire sample is allowed to equilibrate with the vacuum inside the mass spectrometer at the beginning of analysis, which requires that the pumps be isolated from the vacuum chamber. Starting at this time, confounding gases will be introduced through undetectable leaks and desorption from the walls of the vacuum chamber housing the mass spectrometer, and sample gas will be consumed by ionization in the ion source and implantation in either the collector or the walls of the vacuum chamber. Because these processes change the gas composition, and therefore both the abundances and the ratios of the noble gas isotopes being measured, noble gas geochemists typically extrapolate the evolving gas signal back to the time of sample inlet—commonly referred to as "time zero"—meaning that the analysis loses statistical power as it continues in time. The advent of multicollection means that isotope ratios could be computed directly at each time point and then themselves extrapolated to "time zero," but so far noble gas geochemists have largely used multicollection simply as a means to ensure that the maximum amount of data can be collected simultaneously for each isotope.

## 2 Isotopx NGX and ATONA amplifier

The Isotopx NGX is a multicollector noble gas mass spectrometer with a Nier-type ion source, a Hall Probe feedback-controlled electromagnet mass analyzer, and a customizable collector block comprising fixed Faraday cup and ion counting electron multiplier detectors. The source sensitivity is approximately $10^{-3}$ A/Torr, the $^{36}$Ar background is approximately $2 \times 10^{-19}$ moles, or $5 \times 10^{-15}$ cc STP, and the rise is approximately $8 \times 10^{-18}$ moles, or $2 \times 10^{-13}$ cc STP $^{40}$Ar per minute. The NGX at LDEO has five fixed detectors, four Faraday cups and one electron multiplier, in the appropriate configuration to simultaneously collect the five isotopes of argon typically measured for $^{40}$Ar/$^{39}$Ar dating: $^{40}$Ar, $^{39}$Ar, $^{38}$Ar, $^{37}$Ar, and $^{36}$Ar. The electron multiplier is placed at the $^{36}$Ar position, where signals are typically relatively small and must be measured with high precision due to the need for an accurate $^{40}$Ar/$^{36}$Ar ratio for initial Ar correction. We chose this configuration before the ATONA became available, and in fact we believe that an ATONA would be appropriate for $^{36}$Ar measurement in many situations. An instrument with the ability to switch between measuring $^{36}$Ar on an ATONA an an electron multiplier would be able to take advantage of the stability and dynamic range of the ATONA for large $^{36}$Ar signals while still using an ion-counting electron multiplier for very small signals. For example, a single heating step on a very young basalt sample may yield $10^{-14}$ moles of $^{40}$Ar, of which 95% is non-radiogenic. In this case, the uncertainty of the $^{36}$Ar measurement will dominate the trapped Ar correction to the $^{40}$Ar

and therefore the age uncertainty, and we would choose to measure the $3 \times 10^{-17}$ mole $^{36}$Ar signal with the ion counter with

0.2% uncertainty rather than using the ATONA with 3% uncertainty.

After initial installation in late 2017 with Isotopx $10^{11}$ Ω and $10^{12}$ Ω Xact amplifiers, we installed a prototype set of ATONA amplifiers on the NGX in March 2018. The ATONA is a capacitive transimpedance amplifier, which is partially described in UK patent GB2552232. The remaining aspects of the amplifier are protected as trade secrets. The ATONA substitutes the

typical high-gain resistor of an RTIA, for which one would try to minimize the capacitance of the circuit, with a capacitor and a series of proprietary circuits that allow the rate of charge accumulation (rather than the accumulated charge itself) to be continuously sampled (again, the exact mechanism used is a trade secret). Because the ATONA relies on a measurement of the rate of charge accumulation, it simply discharges the feedback capacitor when the rated capacitance has been reached in a process that is transparent to the measurement itself. The proprietary paraelectric dieletric material minimizes nonlinearity due

to current leakage and dielectric hysteresis. Because the Faraday buckets are directly connected to the input of the inverting amplifier, the voltage of the bucket is fixed at zero volts regardless of the accumulated charge on the capacitor and therefore charge buildup that might affect ion behavior is avoided. The result is that the ATONA can measure a wide range of ion beam currents, from attoamps to nanoamps (hence the name), with good linearity, very low noise, and a settling time short enough to be insignificant (less than the 2 ms sampling time of the measurement electronics).

The ATONA has the important characteristic that the noise scales inversely with time, rather than with the square root of time, so accumulating a signal for longer between sampling intervals will result in a linearly less noisy signal. Counting statistics reduces uncertainty with the square root of time, so by comparison the ATONA gains an additional factor of the square root of time in noise reduction when the sampling interval is extended. There is a trade-off in noble gas mass spectrometry because

of the evolution of the signal with time, although it is important to mention that the signal from the production version of the ATONA can be subsampled without sacrificing the gain of the longer sampling time. This dynamic opens up a wide array of possibilities of best measurement practice that will vary with ion beam size, and we have not yet fully explored them; for example, one might choose a longer integration time for smaller beams that are measured as an average and a shorter integration time for larger beams during the same measurement. The work presented here has led us to settle on an integration time of 10

seconds, with a typical total analysis time of 600 seconds in multicollection mode, as a sweet spot for reducing noise without sacrificing gas evolution fit statistics. Analytical conditions for different experiments in this study vary and are described in the figure captions. All isotope evolutions are fit using a linear regression with no outlier data points excluded from either fits or uncertainty calculations, and with no measurement cycles discarded from the analysis. The only exception is in Section 3.3, in which we removed the final 200 seconds from a set of 600-second APIS analyses in order to allow a direct comparison to a

dataset of 400-second analyses on a different mass spectrometer.

## 3 Analyses of electronics and gas standards

### 3.1 Background noise

Reported detector signal units are an arbitrary choice in mass spectrometry; the important quantity for a given detector is signal/noise ratio produced by a given incident ion beam. We quantify this by converting measured signal from detector units to incident ion beam current using Ohm's Law for voltage measured on an RTIA. The ATONA does not measure voltage in the same way as an RTIA, but its firmware converts the signal to equivalent $10^{11}$ $\Omega$ RTIA volts. We convert back to beam current for clearer comparison with RTIAs that have a different gain, and with other types of detectors. As an example, 1 $10^{11}$ $\Omega$ RTIA volt is equivalent to $10^4$ fA, and 625 cps on an ion counting electron multiplier is equivalent to 0.1 fA. We calculate background noise for ideal RTIAs with a variety of feedback resistors. In this case, we assume that the only significant component of noise is Johnson–Nyquist (J–N) noise, or thermal white noise, which is an inherent property of all conductors. The observed noise is caused by the the movement of charge within the conductor in response to random fluctuation caused by thermal radiation, as described by Nyquist (1928) (See Appendix A for equation). J–N noise provides an absolute limit for the signal/noise ratio achievable with an RTIA, and the best commercial RTIAs approach this limit.

Unlike J–N noise, kTC noise (capacitor thermal noise, equal to the product of the Boltzmann constant, k, and the absolute temperature, T, divided by the capacitance, C) has no frequency component. This means that the voltage noise produced by a current discharged from a capacitor will scale linearly with time. As a result, one might expect to achieve a factor of $1/\sqrt{t}$ in noise reduction by extending the charge accumulation time arbitrarily. This is not exactly how the ATONA functions, as one is able to subsample the measurement without losing the benefit of a longer integration time, but the expected linear relationship is achieved, similar to previous systems in which the charge of the capacitor is read directly (Ireland et al., 2014). The theoretical noise floor of the ATONA design is not immediately apparent from the publicly-available information about its capabilities, which do not reveal either the design of the measurement circuit or the value of the capacitor employed. A simple calculation assuming kTC noise is the only source of noise on each ATONA measurement yields a value of 15–20 pF for the complete circuit, which includes both the capacitor used on the amplifier and the capacitance of the Faraday collectors themselves and the wires and feedthroughs that connect them. We measure noise directly through a series of measurements on the Isotopx NGX with the instrument under vacuum, all lenses active, and the filament powered off. We then express this noise floor in terms of incident ion beam for direct comparison to RTIAs.

The results are shown in Table 1 and are plotted in two different ways. First, we show a series of measurements of ATONA noise compared to ideal RTIA J–N noise calculations for a series of RTIA resistor values in Figure 3 (see Appendix A). This figure simply shows measurements taken with the ATONA with no ion beam, with the arithmetic mean of the measurements subtracted from each. This is, therefore, what a series of measurements of a stable beam would look like to the user during a measurement cycle. Each measurement is made with a ten second integration, which is the typical integration time we use for the ATONA on most samples. The ATONA measurements have a standard deviation of 0.0085 fA, which is equivalent to 0.85

$\mu$V on a $10^{11}$ RTIA. In Figure 4, we show the same noise data as 1-$\sigma$ standard deviation of a signal plotted as a function of integration time to show the different behavior of the ATONA as integration time is changed. Using a one second or 100 second integration time, the ATONA measurements have standard deviations of 0.073 fA and 0.0018 fA, respectively. The ten-second integration time value compares favorably to a $10^{13}$ RTIA at 0.011 fA, but does not quite reach the low noise level of a $10^{14}$ RTIA at 0.0040 fA. Similarly, at one second integration, the ATONA is in between the $10^{12}$ and $10^{13}$ RTIA (0.40 fA, 0.037

fA, respectively).

## 3.2 Air standards

We prepared a large air standard of approximately $8.5 \times 10^{-13}$ moles of Ar per aliquot for mass spectrometer installation and initial testing. We used air taken at a distance from the Lamont-Doherty Earth Observatory Comer geochemistry building in Palisades, NY, on a dry day in November, and we filled the approximately six liter standard tank with one aliquot from the

170 approximately 0.1 cc pipette. Subsequent aliquots for measurement were taken from the standard tank using the same pipette, attached to a custom-built high vacuum system containing a hot SAES St101 getter. No primary volume calibration was performed on the pipette for the large standard, so the size of the Ar aliquot was first roughly estimated from the approximate volumes of the standard tank, pipette, and vacuum system, then calculated using intercalibration with a second standard tank with a manometrically-calibrated pipette volume.

We measured four different splits of the air standard ranging from the full aliquot ($8.5 \times 10^{-13}$ moles $^{40}$Ar) to approximately 0.36% of the total ($3.1 \times 10^{-15}$ moles $^{40}$Ar). The split sizes of 100%, 17.7% ($1.5 \times 10^{-13}$ moles $^{40}$Ar), and 2.6% ($2.2 \times 10^{-14}$ moles $^{40}$Ar) are most useful for comparing the Isotopx Xact RTIA to the ATONA. For all Xact measurements, a $10^{11}$ $\Omega$ amplifier was used for $^{40}$Ar and a $10^{12}$ $\Omega$ amplifier was used for $^{38}$Ar. The ATONA amplifiers all use the same feedback

capacitor and are therefore interchangeable. The $^{40}$Ar/$^{38}$Ar ratios for these standards, which provide a direct comparison of the performance of the amplifiers without the effect of the ion counting multiplier used to measure $^{36}$Ar, are shown in Figure 5. For the different shot sizes, the Xact amplifiers produced standard deviations of 0.43%, 3.07%, and 27.9%, respectively, while the ATONA amplifiers produced standards deviations of 0.21%, 1.35%, and 7.87%. As predicted based on zero-beam noise measurements, the ATONA outperforms the Xact for all signal sizes. The improvement between the Xact and the ATONA is

greater for smaller beam sizes because the effect of amplifier J–N noise on the total uncertainty comes to dominate over other factors like source instability when the signal is smaller.

In order to provide a more rigorous assessment of the ATONA amplifiers themselves and to produce an amplifier-only dataset for $^{40}$Ar/$^{36}$Ar, which is a more commonly discussed isotope ratio in $^{40}$Ar/$^{39}$Ar geochronology, we then switched to single col-

190 lector mode. Using the ATONA amplifiers, we measured each species by peak-hopping on the H2 collector, which is normally used for $^{40}$Ar, and we measured $^{40}$Ar/$^{38}$Ar and $^{40}$Ar/$^{36}$Ar for splits of our air standard ranging from 200% (representing two aliquots of the full standard, $1.7 \times 10^{-12}$ moles of Ar, or a beam of approximately 14400 fA) to 0.36%, representing three splits with the extraction line, or approximately $3 \times 10^{-15}$ moles of Ar and a beam of 25.9 fA. The $^{40}$Ar/$^{36}$Ar ratios for these

measurements are shown in Figure 6, and the $^{40}$Ar/$^{38}$Ar ratios are shown in Figure A1. The measured ratios along with internal uncertainties and standard deviations between analyses are shown in Table 2 for $^{40}$Ar/$^{36}$Ar, and in Table A1 for $^{40}$Ar/$^{38}$Ar.

Finally, we measured the same ion beam ($^{40}$Ar) repeatedly on each Faraday detector to determine the gain bias between the different ATONA amplifiers. Choosing the axial detector as a reference, the relative gains of the other detectors ranged was between 1.6 and 3.6 ‰ lower, with a standard deviation of between 106 and 220 ppm for the intercalibration factor of each detector when measured using 1-second integration periods for 10 periods of 10 seconds on each detector (Figure A2). Because we used a real Ar beam measured with a sequential peak hop rather than a synthetically produced calibration voltage, fluctuations in the ion source and mass analyzer electronics might also contribute noise to these measurements, so this is a maximum estimate of the intercalibration drift of the ATONA. The production model of the ATONA amplifiers, which are now being installed on some TIMS instruments, have a calibration voltage that eliminates these other sources of uncertainty; preliminary results from this system show a standard deviation of only 0.6 ppm for each detector when measured using two-minute integration periods over multiple four hour blocks (Szymanowski and Schoene, 2019).

Because the uncertainty of the measured signals is dominated by the thermal noise of the Faraday amplifier, the uncertainty of each measured ratio is controlled largely by the uncertainty of the smaller isotope. For comparison to other instruments, we plot each measured isotope ratio as a function of the sample size of the small isotope in the ratio in Figure 7 (that is, for the same air standard, the $^{40}$Ar/$^{36}$Ar ratio will plot approximately five times higher in terms of sample size than the $^{40}$Ar/$^{38}$Ar ratio, because the $^{40}$Ar/$^{36}$Ar ratio of air is 298.56 while the $^{40}$Ar/$^{38}$Ar ratio of air is 1583.87 (Lee et al., 2006; Mark et al., 2011). This reference frame allows us to compare unlike detectors such as analog multipliers and Faraday cups, as well as to compare isotope ratios measured using a mix of detector types, such as the $^{40}$Ar/$^{36}$Ar ratios measured in the standard multicollection mode of our NGX. While a better reference frame for direct comparison of detector technologies might be beam size rather than sample size, the latter choice allows for a more realistic comparison of mass spectrometers as they are used in the laboratory. We also note that while most noble gas mass spectrometers provide a similar specification for constant pressure ion source sensitivity, field reports indicate that some (notably the Thermo Argus) have an advantage due to both smaller volume and higher constant pressure sensitivity. These results show a clear improvement for the NGX with ATONA compared to the previous generation of mass spectrometer (represented by the LDEO VG 5400) and the NGX with XAct $10^{12}$ $\Omega$ RTIA (the same NGX at LDEO, with its original amplifiers). The performance is also better than published data for the Thermo Argus with $10^{12}$ $\Omega$ RTIA (Mark et al., 2009), despite the Argus' apparently higher source sensitivity, which is consistent with the prediction that the ATONA will easily outperform a $10^{12}$ $\Omega$ RTIA (Figure 3); see Section 3.3 for a comparison to the Argus with a $10^{13}$ $\Omega$ RTIA. Finally, the NGX using its ion counting multiplier in peak-hopping mode is still able to achieve a much lower noise level for very small samples, comparable to the Nu Noblesse with multiple ion counting multipliers (Jicha et al., 2016), which is also consistent with the predicted noise level of the ATONA. However, these detectors are limited to very small samples; the data points with more than $10^{-17}$ moles of $^{36}$Ar in Figure 7 actually use an ATONA for the $^{40}$Ar beam, but we plot them in the ICM category because the uncertainty of the small isotope controls the uncertainty of the ratio measurement.

### 3.3 APIS cocktail standards

The Argon Intercalibration Pipette System (APIS) is a system designed to provide a portable set of argon gas standards of different size and isotope ratio for a noble gas mass spectrometer (Turrin et al., 2015). The APIS has three standard tanks containing air, a cocktail representing argon with a $^{40}Ar/^{39}Ar$ ratio typical of an irradiated Alder Creek sanidine standard, and a cocktail representing argon with a $^{40}Ar/^{39}Ar$ ratio typical of an irradiated Fish Canyon Tuff sanidine standard. Each tank has three pipettes attached to it, with volumes of 0.1, 0.2, and 0.4 cc, allowing aliquots of gas ranging in size from 1 to 7 times the

size of the 0.1 cc pipette to be extracted without resorting to multiple aliquots from a single pipette. We measured each possible size, 0.1 cc, 0.2 cc, 0.3 cc, 0.4 cc, 0.5 cc, 0.6 cc, and 0.7 cc, three times from each of the Alder Creek and Fish Canyon Tuff tanks, and six times from the APIS air standard tank, interspersed with the lab air standard described earlier and procedural blanks.

The APIS standards have accumulated air background since the system was first deployed, so a direct comparison of measured ratios between labs is not possible. However, we can compare air-corrected values for the Fish Canyon and Alder Creek standard tanks—similar to what would be measured during an actual experiment. As an example, we plot measured radiogenic $^{40}Ar*/^{39}Ar$ values ($^{40}Ar/^{39}Ar$ ratios corrected for air contamination using simultaneously measured $^{40}Ar*/^{36}Ar$ ratios) for the Fish Canyon analog from the Isotopx NGX with the ATONA (10-second integration periods; 400 seconds measurement time)

and the Thermo Argus with the $10^{12} \, \Omega$ and $10^{13} \, \Omega$ RTIA (1-second integration periods; 400 seconds measurement time; Figure 8; Ross and Mcintosh, 2016)). While the ATONA exhibits lower noise on a per-signal basis, the higher sensitivity of the Argus ion source makes the results indistinguishable.

### 4 Summary

The ATONA amplifier represents a significant step forward in Faraday cup amplifier technology for noble gas mass spectrom-

250 etry. The ATONA allows a greater dynamic range of ion beams to be measured compared to existing RTIA technology, and only highly specialized RTIA electronics are able to compete with the low noise of the ATONA. The amplifiers are significantly more stable and have higher dynamic range than ion-counting electron multipliers. Other types of mass spectrometer that produce a stable ion beam are likely to see an even greater performance improvement with the ATONA because of its ability to capitalize on long integration times to reduce noise. The strengths of the ATONA, combining low noise for small

samples with high dynamic range and good stability for large samples, are in harmony with the current priorities of the field of noble gas geochemistry, which require instruments that can deliver both high precision and flexibility for measuring a wide range of sample types.

*Author contributions.* SEC installed the instrument, set up hardware and software, designed experiments, performed measurements, and interpreted the results. SRH designed experiments, performed measurements, and interpreted results. DT designed and implemented the

ATONA system and installed the prototype on the NGX at LDEO.

*Competing interests.* DT is an employee of Isotopx, Ltd.

*Acknowledgements.* We thank Daniel Wielandt, Trevor Ireland, and Klaudia Kuiper for particularly thoughtful reviews that contributed greatly to the value of the manuscript. We thank Jake Ross for extensive discussion and Pychron programming help and for sharing raw data from NMGRL measurements published in conference proceedings and Brian Jicha for discussion and for sharing raw data from WiscAr

measurements published in *Chemical Geology*. We also thank Chris Varden for initial installation and testing of the NGX at LDEO.

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

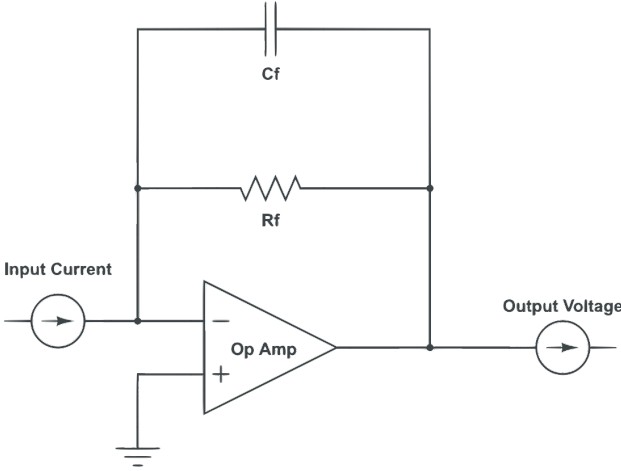

**Figure 1.** Schematic of a transimpedance amplifier. Note that practical examples are far more complex. The circuit consists of an op-amp, which is the active element that converts the input current to a proportional output voltage, and then a feedback resistor and capacitor that determine the gain of the circuit. In a traditional resistance transimpedance amplifier, the resistor is very high value and the capacitance is reduced as much as is practical. The ATONA instead uses a defined capacitance as the feedback element.

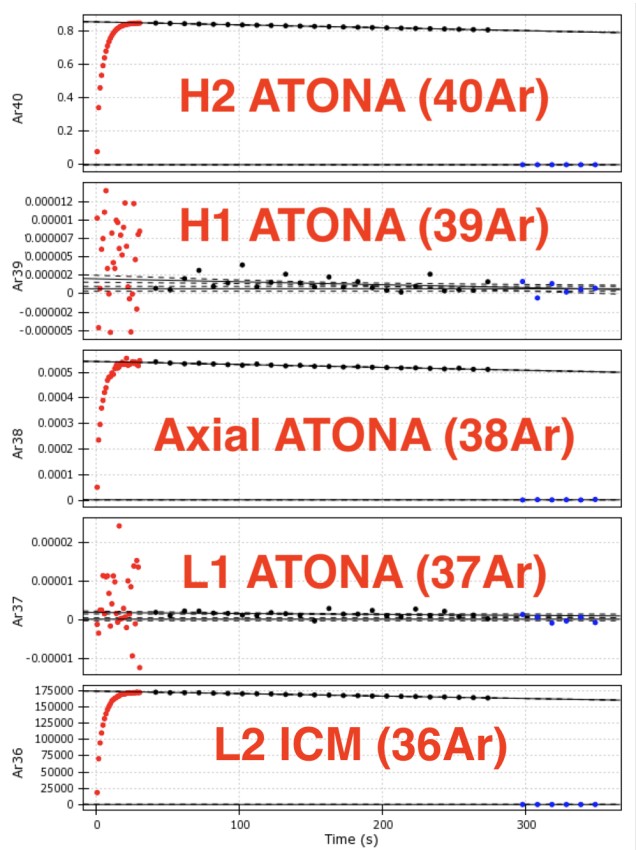

**Figure 2.** An example measurement of the $8.5 \times 10^{-13}$ mole $^{40}$Ar air standard on the Isotopx NGX. The gas is measured with a one-second integration time during 30 seconds of sample inlet, then with a ten-second integration time during 240 seconds of measurement and 60 seconds of baseline measurement. The figure shows the live measurement screen displayed in Pychron during automated sample analysis with overlain labels. Each signal is displayed as reported by the Isotopx software: volts for the four Faraday collectors (converted by the onboard ATONA firmware to equivalent $10^{11}$ Ω RTIA volts) and counts per second for the ion counting electron multiplier.

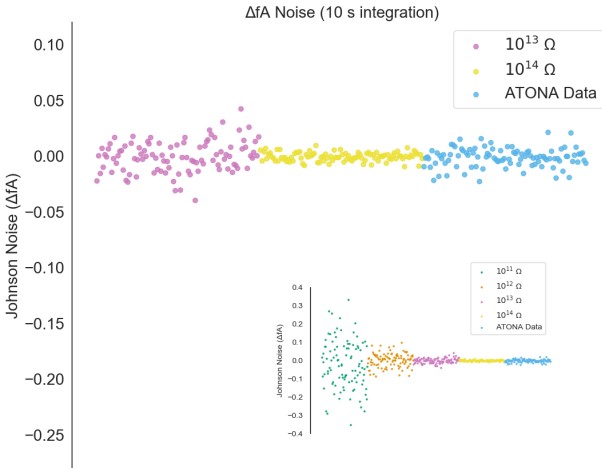

**Figure 3.** Measured noise on the ATONA amplifiers with 10 s integration periods expressed as deviations from the average signal with no ion beam in the mass spectrometer compared to ideal $10^{13}$ $\Omega$ and $10^{14}$ $\Omega$ RTIA noise. The signals are converted to equivalent ion beam current (see Section 3.1. Each measurement and ideal RTIA calculation is made over ten seconds of integration and then simply plotted in order. The inset includes $10^{11}$ $\Omega$ and $10^{12}$ $\Omega$ examples as well, with the same ATONA data. Examples using 1-second and 100-second integration times are included in the appendix (Figures A3 and A4), as is the full version of the inset (Figure A5).

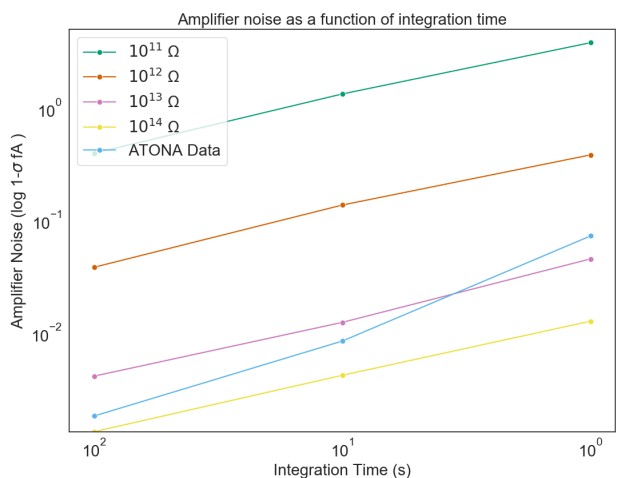

**Figure 4.** Noise expressed as standard deviation of signal measured (ATONA) or calculated (ideal RTIA) for the ATONA and $10^{11}$ Ω, $10^{12}$ Ω, $10^{13}$ Ω, and $10^{14}$ Ω RTIA. The ATONA noise decreases more quickly with increasing integration time because of the $1/t$ (rather than $1/\sqrt{t}$) relationship between noise and integration time.

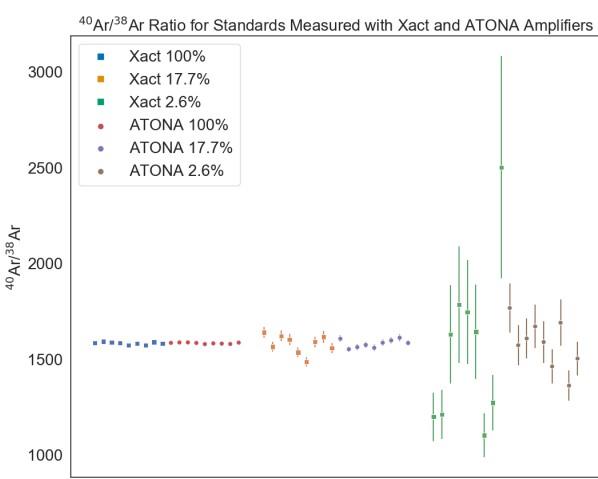

**Figure 5.** $^{40}$Ar/$^{38}$Ar ratios for air standard splits of 100%, 17.7%, and 2.6% measured using both the Isotopx Xact ($10^{11}$ $\Omega$ for $^{40}$Ar and $10^{12}$ $\Omega$ for $^{38}$Ar) amplifiers and the ATONA amplifiers in multicollection mode. Each sequence shows isotope ratios calculated from blank-corrected ratios of extrapolated peak heights for nine air standards measured sequentially, interspersed with blanks between each standard, for 600 seconds each. The ATONA measurements and the Xact measurements were both made using 600 1-second integration periods; ATONA performance improves even further with longer integration periods.

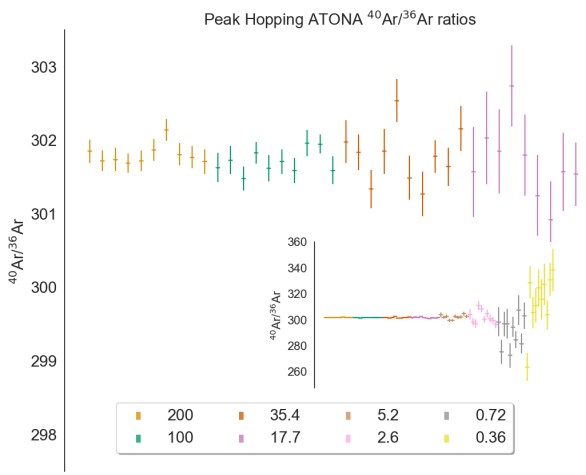

**Figure 6.** $^{40}$Ar/$^{36}$Ar ratios for air standard splits from 200% to 17.7% (inset: 200% and 0.36%) measured using the Isotopx ATONA amplifiers in single collector peak-hopping mode, with the $^{36}$Ar, $^{38}$Ar, and $^{40}$Ar beams measured in sequence on the H2 Faraday. Each beam was measured in sets of three 10-second integration periods, which repeated ten times. Isotope ratios are calculated from blank-corrected ratios of extrapolated peak heights. Each sequence shows ten air standards, plotted interspersed for comparison.

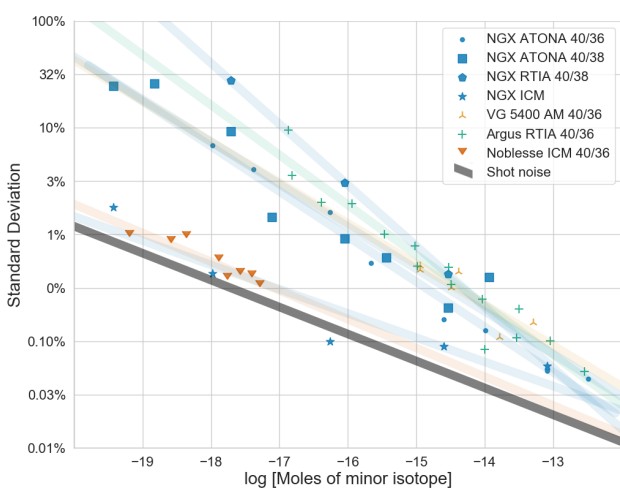

**Figure 7.** Standard deviation of measured $^{40}Ar/^{36}Ar$ or $^{40}Ar/^{38}Ar$ ratios for air standards measured on different mass spectrometers as a function of small isotope abundance in moles (see Section 3.2 for description of data sources). Isotope ratios are calculated from blank-corrected ratios of extrapolated peak heights. The shaded lines are linear fits to each dataset, included primarily as a visual guide. ATONA is the ATONA amplifier described here, RTIA is a traditional resistor transimpedance Faraday amplifier, ICM is an ion counting multiplier, and AM is an analog multiplier. The NGX data points with more than $10^{-17}$ moles of $^{36}Ar$ use an ATONA for the $^{40}Ar$ beam, but in all cases the uncertainty of the small isotope controls the uncertainty of the ratio. This plot provides a direct comparison of whole instrument performance rather than detector performance because the ion source and mass analyzer also contribute to uncertainty in the measurements, and the sample abundance is not weighted by source sensitivity. We note that we are not able to completely control for the effects of different analytical conditions, including background, detector integration time, total measurement time, sensitivity, and data reduction. The limit of shot noise, or counting noise, is shown in grey assuming no other sources of uncertainty and a regression through 600 seconds of analysis. The uncertainty of all detectors will approach this limit at large signals. Note that the uncertainty of the regression is approximately twice the uncertainty one would calculate from an average over the same interval in a mass spectrometry system without an evolving signal.

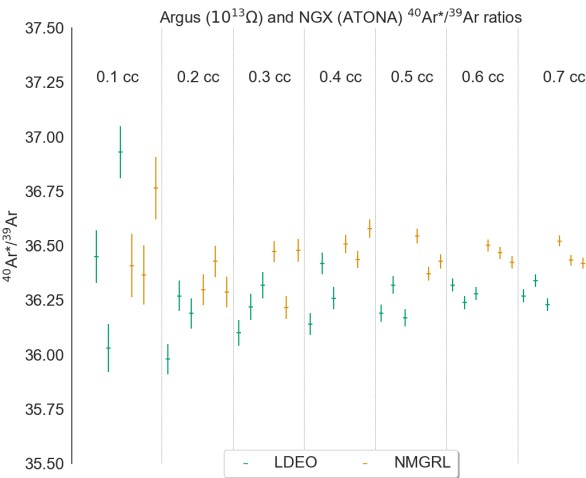

**Figure 8.** Air-corrected $^{40}$Ar*/$^{39}$Ar ratios for the APIS Fish Canyon Tuff analog standard on the Isotopx NGX with the ATONA at Lamont-Doherty Earth Observatory and the Thermo Argus with the $10^{12}$ $\Omega$ and $10^{13}$ $\Omega$ RTIA at the New Mexico Geochronology Research Laboratory (Ross and Mcintosh, 2016), with smaller (0.1 cc) aliquots on the left, and 0.2, 0.3, 0.4, 0.5, 0.6, and 0.7 cc aliquots to the right. Both sets of measurements were performed with 400 seconds of analysis time in multicollection. Isotope ratios are calculated from blank-corrected ratios of extrapolated peak height, with the $^{40}$Ar* corrected for air contamination using the measured $^{36}$Ar. By design, the APIS experiments were conducted according to the same blank and standard protocols in each lab. The ATONA data were collected using 10-second integration periods, while the Argus data were collected using 1-second integration periods. The standard deviation of the signals for a given size aliquot is comparable for the two instruments.

**Table 1.** Standard deviation of the background noise (in fA) for ideal RTIAs and actual standard deviation for measurements for the ATONA with no ion beam (also in fA). A 1 fA beam would produce 0.1 mV on a $10^{11}$ $\Omega$ RTIA.

|  | $10^{11}$ $\Omega$ RTIA | $10^{12}$ $\Omega$ RTIA | $10^{13}$ $\Omega$ RTIA | $10^{14}$ $\Omega$ RTIA | ATONA |
|---|---|---|---|---|---|
| 1 second | 0.4065498 | 0.12983724 | 0.04049524 | 0.01278319 | 0.07310124 |
| 10 second | 0.12743643 | 0.04051549 | 0.01269869 | 0.00408956 | 0.00850963 |
| 100 second | 0.04053762 | 0.01276568 | 0.00407741 | 0.00128429 | 0.00183472 |

## Appendix A: Johnson–Nyquist Noise Calculation

Thermal Johnson–Nyquist noise (J–N noise) is described by Equation 4 from Nyquist (1928):

$$V^2 = 4RK_BT, \tag{A1}$$

where $V$ is the voltage at the frequency of interest, $R$ is the resistance of the circuit, $K_B$ is the Boltzmann constant, and T is the temperature. We rearrange this to solve for voltage noise and then divide by the resistance of the circuit to arrive at the noise fluctuations in terms of beam current $I$.

$$\sigma_I = \sqrt{4RK_BT}/R \tag{A2}$$

This equation is the basis for the calculations shown in Figures 3, 4, A3, A4, A5, A6, and A7.

**Table 2.** $^{40}$Ar/$^{36}$Ar of air standards measured in peak-hopping mode on a single ATONA Faraday cup on the LDEO NGX. Standards were measured using ten cycles each of three ten-second integration periods for each isotope, in the order $^{36}$Ar, $^{38}$Ar, $^{40}$Ar. Uncertainties are 1-$\sigma$ standard deviation.

| Split Size | 200% | | 100% | | 35.4% | | 17.7% | | 5.2% | | 2.6% | | 0.72% | | 0.36% | |
|---|---|---|---|---|---|---|---|---|---|---|---|---|---|---|---|---|
| Moles | 1.7E-12 | | 8.5E-13 | | 3.0E-13 | | 1.5E-13 | | 4.4E-14 | | 2.2E-14 | | 6.1E-15 | | 3.1E-15 | |
| $^{36}$Ar (fA) | 48 | | 25 | | 8.6 | | 4.2 | | 1.2 | | 0.60 | | 0.18 | | 0.081 | |
| | $^{40}$Ar/$^{36}$Ar | ± | $^{40}$Ar/$^{36}$Ar | ± | $^{40}$Ar/$^{36}$Ar | ± | $^{40}$Ar/$^{36}$Ar | ± | $^{40}$Ar/$^{36}$Ar | ± | $^{40}$Ar/$^{36}$Ar | ± | $^{40}$Ar/$^{36}$Ar | ± | $^{40}$Ar/$^{36}$Ar | ± |
| | 301.85 | 0.16 | 301.63 | 0.2 | 301.98 | 0.29 | 301.57 | 0.61 | 303.95 | 1.95 | 304.18 | 4.07 | 298.44 | 11.21 | 263.6 | 10.83 |
| | 301.72 | 0.14 | 301.73 | 0.19 | 301.84 | 0.24 | 302.03 | 0.63 | 301.91 | 1.17 | 298.3 | 2.88 | 275.24 | 9.29 | 328.75 | 12.51 |
| | 301.74 | 0.16 | 301.48 | 0.16 | 301.34 | 0.26 | 301.85 | 0.57 | 302.75 | 1.39 | 297.1 | 3.14 | 296.75 | 9.44 | 305.53 | 11.93 |
| | 301.69 | 0.13 | 301.83 | 0.15 | 301.85 | 0.31 | 302.74 | 0.55 | 299.69 | 1.31 | 310.96 | 3.49 | 296.99 | 11.28 | 311.06 | 13.04 |
| | 301.72 | 0.14 | 301.62 | 0.18 | 302.54 | 0.29 | 301.8 | 0.55 | 299.5 | 0.99 | 308.39 | 2.59 | 272.54 | 9.51 | 324.64 | 14.03 |
| | 301.87 | 0.15 | 301.71 | 0.17 | 301.49 | 0.3 | 301.25 | 0.55 | 302.6 | 1.32 | 300.24 | 2.4 | 294.46 | 7.82 | 315.81 | 15.25 |
| | 302.14 | 0.15 | 301.59 | 0.17 | 301.27 | 0.3 | 300.92 | 0.52 | 301.75 | 1.21 | 304.97 | 3.17 | 284.62 | 6.65 | 327.28 | 15.75 |
| | 301.81 | 0.15 | 301.96 | 0.18 | 301.78 | 0.22 | 301.57 | 0.53 | 302.38 | 1.29 | 300.98 | 3.18 | 307.42 | 11.47 | 303.86 | 11.31 |
| | 301.77 | 0.15 | 301.95 | 0.13 | 301.64 | 0.26 | 301.54 | 0.43 | 304.76 | 1.52 | 299.41 | 2.62 | 281.71 | 7.78 | 330.93 | 12.68 |
| | 301.71 | 0.17 | 301.59 | 0.19 | 302.16 | 0.31 | 301.58 | 0.49 | 302.64 | 1.41 | 296.59 | 3.09 | 303.07 | 9.99 | 338.41 | 16.31 |
| Average | 301.8 | | 301.7 | | 301.8 | | 301.7 | | 302.2 | | 302 | | 291 | | 315 | |
| SD (1-$\sigma$) | 0.13 | | 0.16 | | 0.38 | | 0.48 | | 1.6 | | 4.9 | | 12 | | 21 | |
| SD (1-$\sigma$ %) | 0.044% | | 0.053% | | 0.13% | | 0.16% | | 0.54% | | 1.6% | | 4.1% | | 6.8% | |

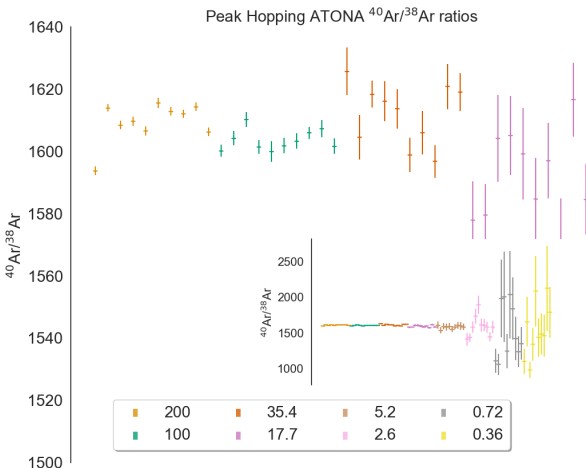

**Figure A1.** $^{40}$Ar/$^{38}$Ar ratios for air standard splits from 200% to 17.7% (inset: 200% and 0.36%) measured using the Isotopx ATONA amplifiers in single collector peak-hopping mode, with the $^{36}$Ar, $^{38}$Ar, and $^{40}$Ar beams measured in sequence on the H2 Faraday. Each beam was measured in sets of three 10-second integration periods, which repeated ten times. Each sequence shows ten air standards, plotted interspersed for comparison.

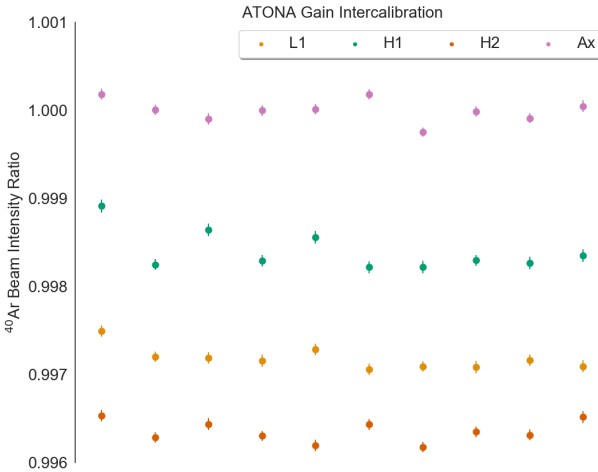

**Figure A2.** Intercalibration measurements using an $^{40}$Ar beam produced by aliquots of the $8.5 \times 10^{-13}$ mole air standard, measured by peak hopping just the $^{40}$Ar beam on each of the four ATONA Faraday collectors on the NGX. Plotted are the ratios of each measurement of the $^{40}$Ar signal on a given detector to the average of all measurements on the Axial detector. Measurements were made using sets of ten 1-second integration periods, repeated ten times sequentially on each detector, with the intensities calculated using a linear extrapolation to time-zero; internal uncertainties shown are the $1 - \sigma$ standard error of the linear fit. No blank correction was made. The detector intercalibration factor ranges from 0.9964 to 0.9984 for the other three detectors relative to the axial detector, with standard deviations ranging from 106 to 220 ppm for each.

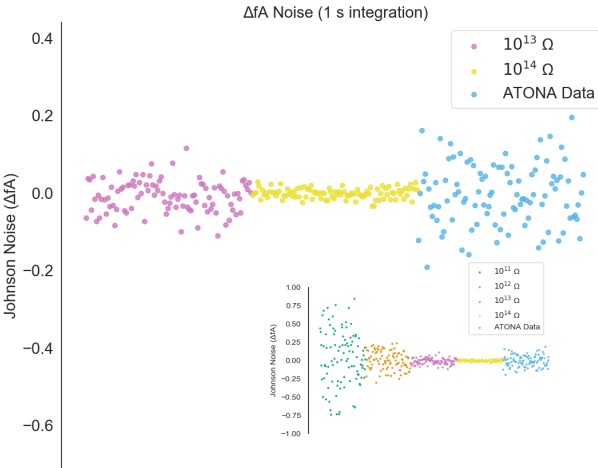

**Figure A3.** Measured noise on the ATONA amplifiers with 1 s integration periods expressed as deviations from the average signal with no ion beam in the mass spectrometer compared to ideal $10^{13}$ Ω and $10^{14}$ Ω RTIA noise. The signals are converted to equivalent ion beam current (see Section 3.1. Each measurement and ideal RTIA calculation is made over one second of integration and then simply plotted in order. The inset includes $10^{11}$ Ω and $10^{12}$ Ω examples as well, with the same ATONA data. Examples using a 10-second integration times are included in Figure 3. The full version of the inset is provided in Figure A6.

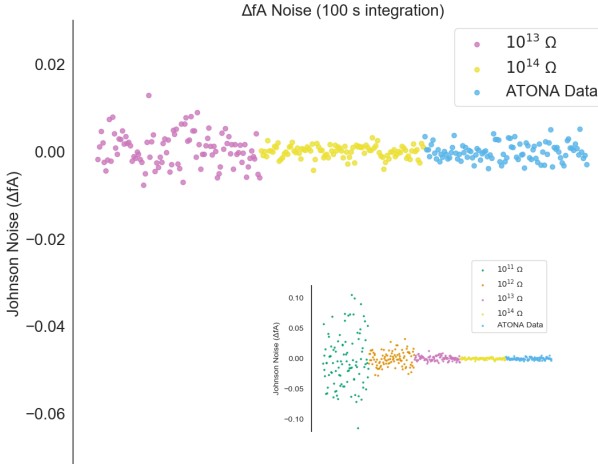

**Figure A4.** Measured noise on the ATONA amplifiers with 100 s integration periods expressed as deviations from the average signal with no ion beam in the mass spectrometer compared to ideal $10^{13}$ $\Omega$ and $10^{14}$ $\Omega$ RTIA noise. The signals are converted to equivalent ion beam current (see Section 3.1. Each measurement and ideal RTIA calculation is made over 100 seconds of integration and then simply plotted in order. The inset includes $10^{11}$ $\Omega$ and $10^{12}$ $\Omega$ examples as well, with the same ATONA data. Examples using a 10-second integration times are included in Figure 3. The full version of the inset is provided in Figure A7.

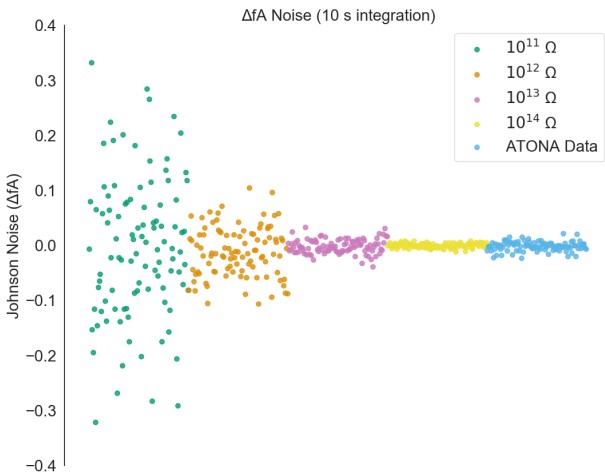

**Figure A5.** Measured noise on the ATONA amplifiers with 10 s integration periods expressed as deviations from the average signal with no ion beam in the mass spectrometer compared to ideal $10^{11}$ $\Omega$, $10^{12}$ $\Omega$, $10^{13}$ $\Omega$, and $10^{14}$ $\Omega$ RTIA noise. The signals are converted to equivalent ion beam current (see Section 3.1. Each measurement and ideal RTIA calculation is made over 10 seconds of integration and then simply plotted in order. This is the full version of the inset from Figure 3.

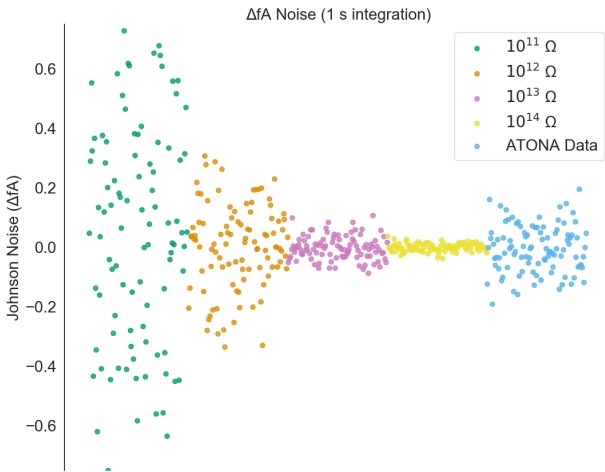

**Figure A6.** Measured noise on the ATONA amplifiers with 1 s integration periods expressed as deviations from the average signal with no ion beam in the mass spectrometer compared to ideal $10^{11}$ $\Omega$, $10^{12}$ $\Omega$, $10^{13}$ $\Omega$, and $10^{14}$ $\Omega$ RTIA noise. The signals are converted to equivalent ion beam current (see Section 3.1. Each measurement and ideal RTIA calculation is made over 10 seconds of integration and then simply plotted in order. This is the full version of the inset from Figure A3.

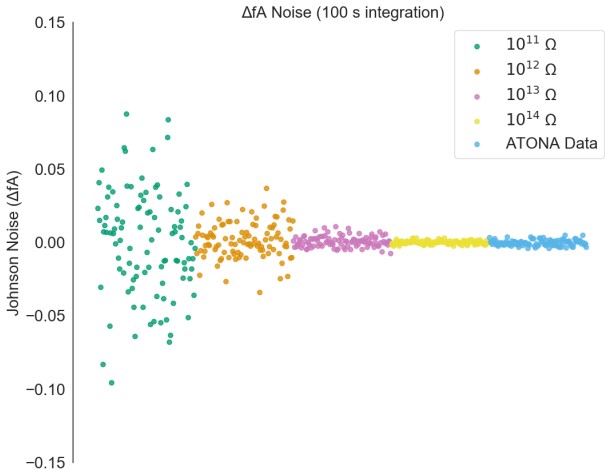

**Figure A7.** Measured noise on the ATONA amplifiers with 100 s integration periods expressed as deviations from the average signal with no ion beam in the mass spectrometer compared to ideal $10^{11}$ $\Omega$, $10^{12}$ $\Omega$, $10^{13}$ $\Omega$, and $10^{14}$ $\Omega$ RTIA noise. The signals are converted to equivalent ion beam current (see Section 3.1. Each measurement and ideal RTIA calculation is made over 10 seconds of integration and then simply plotted in order. This is the full version of the inset from Figure A4.

**Table A1.** $^{40}$Ar/$^{38}$Ar of air standards measured in peak-hopping mode on a single ATONA Farday cup. Standards were measured using ten cycles each of three ten-second integration periods for each isotope, in the order $^{36}$Ar, $^{38}$Ar, $^{40}$Ar. Uncertainties are 1-$\sigma$ standard deviation.

| Split Size | 200% | | 100% | | 35.4% | | 17.7% | | 5.2% | | 2.6% | | 0.72% | | 0.36% | |
|---|---|---|---|---|---|---|---|---|---|---|---|---|---|---|---|---|
| Moles | 1.7E-12 | | 8.5E-13 | | 3.0E-13 | | 1.5E-13 | | 4.4E-14 | | 2.2E-14 | | 6.1E-15 | | 3.1E-15 | |
| $^{38}$Ar (fA) | 9.0 | | 4.4 | | 1.6 | | 0.79 | | 0.24 | | 0.12 | | 0.037 | | 0.018 | |
| | $^{40}$Ar/$^{38}$Ar | ± | $^{40}$Ar/$^{38}$Ar | ± | $^{40}$Ar/$^{38}$Ar | ± | $^{40}$Ar/$^{38}$Ar | ± | $^{40}$Ar/$^{38}$Ar | ± | $^{40}$Ar/$^{38}$Ar | ± | $^{40}$Ar/$^{38}$Ar | ± | $^{40}$Ar/$^{38}$Ar | ± |
| | 1593.77 | 1.34 | 1600.24 | 1.96 | 1625.76 | 7.67 | 1577.95 | 12.48 | 1606.37 | 51.64 | 1410.08 | 91.6 | 1105.81 | 167.86 | 1097.03 | 175.99 |
| | 1614 | 1.23 | 1604.26 | 2.29 | 1604.56 | 7.16 | 1579.56 | 9.93 | 1533.66 | 43.81 | 1432.79 | 60.02 | 1055.65 | 147.96 | 1656.38 | 348.27 |
| | 1608.49 | 1.33 | 1610.22 | 2.43 | 1618.39 | 4.32 | 1604.27 | 13.97 | 1592.27 | 42.28 | 1581.09 | 77.27 | 1982.04 | 543.76 | 977.93 | 114.47 |
| | 1609.75 | 1.42 | 1601.48 | 2.18 | 1616.19 | 6.44 | 1605.08 | 12.7 | 1579.97 | 50.73 | 1732.52 | 102.8 | 2006.36 | 637.14 | 1337.71 | 230.81 |
| | 1606.59 | 1.49 | 1599.95 | 3.34 | 1613.67 | 6.25 | 1599.19 | 14.71 | 1591.78 | 45.32 | 1897.26 | 127.61 | 1243.5 | 238.71 | 2084 | 494.11 |
| | 1615.55 | 1.67 | 1601.88 | 2.42 | 1598.89 | 5.47 | 1584.81 | 13.06 | 1552.6 | 42.78 | 1610.14 | 89.48 | 2035.09 | 613.68 | 1435.19 | 267.91 |
| | 1612.93 | 1.37 | 1603.37 | 2.53 | 1606.04 | 7.01 | 1597.03 | 12.13 | 1585.02 | 39.16 | 1608.49 | 86.12 | 1841.06 | 439.03 | 1485.93 | 290.36 |
| | 1612.13 | 1.34 | 1605.97 | 1.92 | 1596.85 | 5.24 | 1570.72 | 14.27 | 1605.16 | 44.37 | 1580.82 | 76.43 | 1420.77 | 301.82 | 1462.66 | 298.32 |
| | 1614.38 | 1.33 | 1607.31 | 2.79 | 1620.84 | 7.21 | 1616.69 | 11.86 | 1597.96 | 55.59 | 1442.38 | 63.65 | 1236.64 | 214.53 | 2123.91 | 600.62 |
| | 1606.3 | 1.35 | 1601.69 | 2.44 | 1619.13 | 6.02 | 1584.63 | 11.23 | 1583.43 | 42.14 | 1580.97 | 91.15 | 1352.15 | 232.3 | 1787.73 | 358.78 |
| Average | 1609 | | 1604 | | 1612 | | 1592 | | 1583 | | 1588 | | 1528 | | 1545 | |
| SD (1-$\sigma$) | 6.4 | | 3.3 | | 9.8 | | 15 | | 23 | | 147 | | 394 | | 378 | |
| SD (1-$\sigma$ %) | 0.40% | | 0.21% | | 0.61% | | 0.92% | | 1.5% | | 9.3% | | 26% | | 24% | |