# Peer review of "The Isotopx NGX and the ATONA Faraday Amplifiers"

_Geochronology, 2020_

## Referee Comment (RC1) · Daniel Wielandt (Referee) · 13 Mar 2020

General comments: This is a well-written article on an important novel amplifier technology called ATONA that provides an otherwise currently unavailable combination of low noise and high dynamic range for Faraday cup measurements of ion beams. The technology could significantly improve both current and future mass spectrometers, and is therefore of general interest to all mass spectrometry specialists. I however believe that its impact could be improved by including some additional information as mentioned below. Alternatively, the suggestions in general comments should be addressed in future publications.

The article focuses on comparing the ATONA to current 10E11, 10E12, 10E13 and

a hypothetical 10E14 ohm amplifier, or rather their idealized Johnson-Nyquist noise characteristics, for the purpose of multicollector noble gas measurements. ATONA outperforms ideal i.e. model 10E13 ohm amplifiers with respect to signal-noise ratio for 10 second integrations which is (most likely) an appropriate integration time for many measurements, and approaches an ideal (and currently commercially unavailable) 10E14 ohm amplifier for a 100 second integration which is most likely to long to properly sample and back-project a noble gas ion beam evolution to T0. The high dynamic range and low noise-fast response is definitely an improvement as compared to traditional amplifiers. This versatility means that amplifiers do not need not be physically or electronically switched among Faraday cups for different applications, which is an additional advantage that complements their low-noise characteristics. An ATONA could also be useful for single detector instruments that still have merit due to the high sensitivities afforded by the small volumes of such instruments.

Although the comparison with traditional amplifiers at low signal intensities is appropriate, the paper could benefit from a more stringent comparison with ion counters where the noise characteristics at low signal intensities are dominated by Poisson i.e. counting noise of the individual ion arrivals. This noise is inherent to counting atoms or ions and cannot be avoided. An interesting question is therefore under which beam intensity x time i.e. accumulated charge conditions the "baseline" noise in an ATONA becomes comparable to this inherent and unavoidable counting noise that will also be present and superimposed on zero-beam i.e. electronic baseline noise? This would seems to be an appropriate lower dynamic range where ion counters would (decisively?) outperform ATONA in terms of precision (but not necessarily accuracy). This number could presumably be calculated based on the 1-10-100 second zero-beam measurements that have already been carried out. It may also be possible to tease out that information from e.g. figure 7, but it is better that it is presented.

The paper would also benefit if the working principles of ATONA were more thoroughly discussed (without disclosing confidential information). The patent documents contain

a lot of public information that could be condensed into a description of the technology. I think the mass spectrometry community would be more likely to adopt the technology if they could understand it better, rather than using it as a "black box" technology where one might run into an unpredictable problem. As a naive non-engineer I personally would like to know how leakage current is reduced. Is there a maximum charge than can be accumulated before "discharging" if that is even the appropriate term? Are there hysteresis effects in the capacitor that make it particularly hard to drive out or sense low buildups of charge that might adversely affect linearity at low signal intensities? Can charge buildup in the Faraday-amplifier system start to deflect incoming ions, changing the peak shape thereby affecting e.g. pseudo-resolving peak-shoulder measurements. Does the "firmware" make decisions on sampling rate or readout parameters, switching between different regimes that depend on beam intensities?

Throughout: The term Johnson-Nyquist noise is used in line 114, but then subsequent usage is about Johnson noise. Should abbreviate it JN-noise at first usage, and then refer to it as such subsequently.

When discussing the performance using air and cocktail standards, it would be nice to have the approximate beam intensities tabulated in e.g. fA as that it the unit that is already reported for noise measurements.

Specific: First paragraph i.e. 8-22 could perhaps use a statement regarding engineering tradeoffs regarding multicollection versus volume/sensitivity, i.e. the increase in volume that tends to occur with multicollection and the related drop in sensitivity. This is one reason why single collector instruments still have a role. In fact, the versatility of the ATONA seems to make it very well suited for that role; this is only aided by its rapid response as discussed later.

Second paragraph, line 30. Mention of long settling time for high value resistors is relevant in case of dynamic measurements, but static multi-collection of noble gases all but removes the settling time issue since any single resistor only measures one

very slowly evolving beam. This should be mentioned in order to be fair to the current generation of high-ohm multi-collector equipped instruments.

Paragraph 6, Line 80. Could the authors perhaps make a back of the envelope error propagation calculation of how much of the air correction error on a blank subtraction on their instrument would arise from the 36Ar using a ion counter versus the ATONA? Or conversely the calculations suggested in the general comments regarding comparison of counting noise vs zero beam noise? This would be highly relevant for e.g. Ar or Ne-dating of young samples where samples or fractions may be comparable in intensity to blanks.

Paragraph 7, Line 84. If possible, it would be nice if the patent were hyperlinked.

Paragraph 9 A formulation of Johnson-Nyquist noise with some appropriate reference and description would be useful for non-specialists.

Paragraph 11, Lines 130-140. This is a bit hard to read, and the reporting would benefit from a data-table showing the noise characteristics for 1, 10 and 100 second integrations with ATONA and 10E11-14 resistors. In such a way, one could focus on describing the noise "crossover" points for the various detector technologies that most readers would be searching for anyway as seen in figure 4.

Figure 6 (and figure A1) It is hard to identify the ranges, could the color code somehow be complemented by a change in marker style? It might also be a good idea to write the ranges as from 200% to 0.36% rather than between 200% and 0.36%.

Figure 7 We should expect a number of inflection points where all faraday mass spectrometer technologies gradually switch to follow a slope determined by counting noise (N^0.5) rather than signal over "baseline" JN or kTC noise (N^1). The linear error envelopes could give the erroneous impression that Faraday-based technologies can eventually outperform counting noise at high intensities, this should be avoided.

Table 1 The table should include the intensity of the smallest ion beam intensity i.e. the

36Ar intensity in fA. It would also be nice to have (and discuss) an MSWD to compare internal precision and external reproducibility for all the measurements. A calculated average for the different intensities would also be nice, and could be plotted to evaluate non-linearity.

Table A1 The table should include the intensity of the smallest ion beam intensity i.e. the 38Ar intensity in fA. It would also be nice to have (and discuss) an MSWD to compare internal precision and external reproducibility for all the measurements. A calculated average for the different intensities would also be nice, and could be plotted to evaluate non-linearity.

Data for 0.36% measurements seem improbably precise, are they missing a digit?

It would also be nice to discuss the presumably significant decrease in precision when going from 5.2% aliquots to 2.6% aliquots and lower. Is this a characteristic of ATONA, or is it due to error propagation effects from subtraction of blank 38Ar + H37Cl?

---

## Short Comment (SC1) · 13 Mar 2020

General comments

This article describes the performance of a new patented type of capacitive transimpendence amplifier (CTIA) for noble gas mass spectrometry. Due to trade secrets the exact working of this amplifier is not described, only its performance is tested and compared to other commonly used amplifier technology. This seems to be a new step in amplifier development and although not fully disclosed, this is an development that likely will be implemented by several labs in the next 5 years or so. I therefore consider this paper worth publishing, because it is relevant for the community to judge the possible advantages and disadvantages of this new CTIA. The papers is well written and

clearly describes the experiments and tests that have been performed.

Specific comments and technical issues

Line 38 "as those are that are" → remove "are"

Line 63 "through small leaks". What do you consider small leaks?

Line 77-80: What about 37 beam. This is also a very small beam on e.g. young sanidine grains (can now possible be addressed with ATONA).

Line 96-97: Not fully clear, can you give examples of approaches you are thinking of (even tough not fully tested)?

Line 130-131: Can you provide used equations and calculations in appendix?

Line 142: modify to "approximately $8.5\times10$-13 moles of 40Ar per aliquot"

Line 146-149: Can you quantify? What signals did you expect based on your approximations and based on GLO? What is the 40Ar* content in your GLO standard? And I'll assume you mean APIS with the manometrically calibrated volume. Can you add an estimate of your system's sensitivity?

Line 152: Can you add for clarity 100% ($8.5\times10$-13 moles 40Ar), 37.7 % (x moles 40Ar) etc

Line 211: "our lab standard" which is?

Line 213-214: "so a direct comparison of measured ratios is not possible" Comparison with what?

Line 224-225: "gain bias of the amplifiers is significantly more stable than both RTIAs and electron multipliers" This paper does not really provide data for comparison of gains for RTIAs and ATONA. Only gain data of ATONA are shown.

Figure 1: add the unit between brackets to the Y-axis title (e.g. cps). In caption it is mentioned that Faraday data are reported in Volts, in line 110-111 it is stated that you

**GChronD**
[Figure]

convert back to beam current for easier comparison.

Figure 3 and text line 112-115: can you add your calculations / formulas used for RTIA noise to the appendix. Inset is really small and hard to read (especially when printed)

Figure 5: Is 40Ar/38Ar the t0 intensity of 40Ar air minus t0 intensity of 40Ar blank divided by the t0 intensity of 38Ar air minus t0 intensity of 38Ar blanks? How many blanks are run? In the legend there are only circle symbols, in the figure also squares. The way data are plotted suggests that Xact and ATONA measurements are bracketed. Can you first plot the Xact 100% data, followed by ATONA 100% data etc.?

Figure 6. Maybe a matter of reader preference, but I prefer to see the 10 different analyses of one beam size plotted combined instead of interspersed. Now I find it difficult to see that variation within 10 similar experiments. And we are looking at ratios of blank corrected time zero intensities of 40 and 36? Inset is again rather small.

Figure 7. The ARGUS RTIA data are from NMGRL? And are the measured with m/e36 on a Faraday with RTIA or multiplier? Colors of shaded lines are similar, not clear what they are showing.

Figure 8: in caption it is indicated that smaller aliquots are on the left, and larger on the right. Can you indicate different areas in the figure which are the 0.1cc, the 0.2cc aliquots etc. The NMGRL Argus measurements are with 40Ar on H2 with 10^12 Ohm amplifier and 36 on L2 with 10^13 Ohm amplifier? Do Argus data with 40Ar on H1 with 10^12 or 10^13 Ohm amplifier and 36Ar on L3 multiplier also exist? And if yes, how do they compare? Did NMGRL perform exactly the same experiment with 3 aliquots per pipette volume? And if not, what are the criteria to select these 3 data points?

Table 1: what is the $\pm$ in the header row? 1SD? What is the 1-$\sigma$ at the bottom of the table: the standard deviation of the ten measurements? Can you also report the mean (or the weighted mean)?

Figure A1: I don't like the interspersed way of plotting. Also with all the colors that look

rather similar it is difficult to see what is what.

Figure A2: what is exactly plotted on the Y axis? Why not signal divided by average AX signal? Then the intercalibration factors mentioned in caption are immediately clear. And what is plotted on the X-axis? Why are there no data of aliquots 817, 820 etc. What is the beam size used for this intercalibration, is a baseline correction needed? And I'll assume data are regressed to time zero using a linear fit? What is the settling time, maybe worth mentioning, because a similar approach using RTIAs will take longer.

---

## Referee Comment (RC2) · Trevor Ireland (Referee) · 20 Mar 2020

Referees Comment on ATONA Charge mode system.
Trevor Ireland, RSES ANU.

This is a timely paper concerning the introduction of the ATONA charge mode collection system to an ISOTOPX NGX.

The challenge of getting the the capacitative feedback system to work has been ongoing for a number of years.  Both Cary and Keithley electrometers have had the ability to measure charge accumulation, but getting it to work in a routine analytical setup has been somewhat difficult (see Esat 1995, and Ireland et al. 2014).

The physics behind charge mode and the implications for data collection are interesting, but are not particularly well explained in this paper.  Potentially this is because of the patent that is being sought for this system.  In any case, the issues concerning the noise floor for a capacitive system effectively relate to a "read" noise in the capacitor system as opposed to the Johnson Noise in a resistive system.  Hence, there is a constant noise component in the capacitative read system (see Ireland et al. 2014), and the longer you integrate the better the signal to noise. On the other hand, you continually integrate Johnson Noise and so the signal/noise does not improve as quickly for an increase in integration time.  For comparison, we have set up our capacitative system for a 2 s integration and the noise is similar to the 10e-13 ohm resistor, pretty much similar to what is achieved here.  The 2s integration is appropriate for an ion microprobe because of the continual change in analytical conditions.  A longer integration time is fine for a noble gas instrument because the gas is effectively homogenised in the source and there are only longer term fractionation processes to deal with.

The work in the Isotope NGX is based around noble gas analysis, and specifically Ar isotopes.  On one hand, this is a good system to look at because there is a good dynamic range in the isotope ratios under consideration.  It also has the benefit that the ion beam is only changing at a (slow) steady rate allowing a good description of the progression of the counting statistics.  The data show that the system performs well at the level commensurate with the measured ratios.  On the other hand, Ar isotopes are not typically measured to high precision (e.g. as might be achieved for TIMS or ICP-MS analysis, or even SIMS analysis).  This makes it also more difficult to establish the linearity of the system as well.

It is evident that the noise floor is still an issue for the 36Ar measurements described here.  So as the volume of Ar gas is reduced, the error magnification from measuring the 36Ar/40Ar and resulting corrections to 40Ar/39Ar are still going to be a limitation and will likely still need to be carried out on an electron multiplier.

The benefit of the capacitative system is that measurements can be carried out on more Faraday cups, and potentially without the need for an electron multiplier (ion counter).  The noise floor we have achieved is better than 500 c/s which means that for most isotope ratio measurements Charge mode is adequate and very often superior to an electron multiplier.   At the upper level of count rates, 250-1000 x 10e3 c/s, the charge mode seamlessly connects with 10e12 ohm resistor capability.  At the lower end, we have measured isotope ratios down to 10,000 c/s, which is well removed from the gain drift and dead time issues of an electron multiplier.  As such charge mode does provide that connection between electron multipliers and the traditional resistor feedback amplifiers.  But as demonstrated in this paper, it is a complementary aspect of the measurement of isotope ratios in geochemistry.

References
T.M. Esat, Charge collection thermal ionization mass spectrometry of thorium. International Journal of Mass Spectrometry and Ion Processes 148 (1995) 159–170.

T.R. Ireland, N. Schram, P. Holden, P. Lanc, J. Ávila, R. Armstrong, Y. Amelin, A. Latimore, D. Corrigan, S. Clement, J.J. Foster, W. Compston, Charge-mode electrometer measurements of S-isotopic compositions on SHRIMP-SI. International Journal of Mass Spectrometry 359 (2014) 26–37.

---

## Author Comment (AC1) · 6 May 2020

*Klaudia Kuiper k.f.kuiper@vu.nl*

*General comments This article describes the performance of a new patented type of capacitive transimpendence amplifier (CTIA) for noble gas mass spectrometry. Due to trade secrets the exact working of this amplifier is not described, only its performance is tested and compared to other commonly used amplifier technology. This seems to be a new step in amplifier development and although not fully disclosed, this is an development that likely will be implemented by several labs in the next 5 years or so. I therefore consider this paper worth publishing, because it is relevant for the community to judge the possible advantages and disadvantages of this new CTIA. The papers is well written clearly describes the experiments and tests that have been performed.*

*Specific comments and technical issues*

*Line 38 "as those are that are" → remove "are"*

> We have corrected this typo.

*Line 63 "through small leaks". What do you consider small leaks?*

> We have changed the word "small" to "undetectable" to clarify that we mean small inputs of gas that are too small to be considered problematic and would not be detected through leak checking.

*Line 77-80: What about 37 beam. This is also a very small beam on e.g. young sanidine grains (can now possible be addressed with ATONA).*

> Sanidine will have very low 37Ar; the size of this beam will still be far too small to measure precisely with the ATONA. This is especially true for young sanidine, which typically undergoes very short irradiation. The other side of this coin is that the correction is so small that the precision we obtain is acceptable. It is certainly true that we are measuring 37Ar more precisely on the NGX with ATONA than on previous instruments.

*Line 96-97: Not fully clear, can you give examples of approaches you are thinking of (even tough not fully tested)?*

We have added an example of a measurement approach that will be possible with the production version of the ATONA hardware (it is not possible with the protoype version we used here).

*Line 130-131: Can you provide used equations and calculations in appendix?*

We have added the equation to the appendix.

*Line 142: modify to "approximately 8.5×10-13 moles of 40Ar per aliquot"*

We have clarified that the amount of Ar is calculated per aliquot.

*Line 146-149: Can you quantify? What signals did you expect based on your approximations and based on GLO? What is the 40Ar\* content in your GLO standard? And I'll assume you mean APIS with the manometrically calibrated volume. Can you add an estimate of your system's sensitivity?*

I think the original wording of this section makes it sound like something more complicated is happening here. The standard used in this paper was prepared without first doing a manometric volume calibration for the machined pipette, so the point of this section is just to say that the precise size of this particular standard is known from a comparison to properly volume-calibrated air standard in a different tank. APIS is not involved. The calculated sensitivity was also compared favorably to GLO in the course of normal analyses on several occasions, but this is not used as part of the primary calibration and is not particularly important, so I have removed this remark for clarity. Hopefully the changes in the manuscript make all of this more clear. The sensitivity of the mass spectrometer is described elsewhere in the paper.

*Line 152: Can you add for clarity 100% (8.5×10-13 moles 40Ar), 37.7 % (x moles 40Ar) etc*

Yes, we have added this information.

*Line 211: "our lab standard" which is?*

The original wording in the paper is very confusing. The lab standard is the afore-mentioned air standard. The other air standard is the air standard that is part of the APIS. We have clarified this in the manuscript.

*Line 213-214: "so a direct comparison of measured ratios is not possible" Comparison with what?*

We have added "between labs." The original purpose of the APIS experiment was to allow mass spectrometers to be compared after measuring exactly the same gas. While the comparison is still useful, the noticeable amount of air contamination over time requires that the ratios first be corrected before comparison.

*Line 224-225: "gain bias of the amplifiers is significantly more stable than both RTIAs and electron multipliers" This paper does not really provide data for comparison of gains for RTIAs and ATONA. Only gain data of ATONA are shown.*

This is a fair point. We have changed this to focus on noise levels and to clarify the comparison to the ion-counting multiplier. The preliminary data we cite from TIMS instruments (Szymanowski and Schoene, 2019) will show more clearly that the gain stability of the ATONA is superior to existing RTIAs. In our case, we do not have an independent electronic calibration on the prototype unit, and the stability of the signal is limited by the noble gas mass spectrometer ion source rather than by drift in the amplifiers.

*Figure 1: add the unit between brackets to the Y-axis title (e.g. cps). In caption it is mentioned that Faraday data are reported in Volts, in line 110-111 it is stated that you convert back to beam current for easier comparison.*

This figure shows the output as displayed in the current version of the pychron software, which is not how we present the data elsewhere in the paper. For this reason, we clarify the display units in the caption.

*Figure 3 and text line 112-115: can you add your calculations / formulas used for RTIA noise to the appendix. Inset is really small and hard to read (especially when printed)*

The formula has been added to the appendix. I am also adding the full version of the inset figure to the appendix because I do not think it merits another figure in the paper and the template restricts the size of the figures, so it is difficult to make the inset more readable.

*Figure 5: Is 40Ar/38Ar the t0 intensity of 40Ar air minus t0 intensity of 40Ar blank divided by the t0 intensity of 38Ar air minus t0 intensity of 38Ar blanks? How many blanks are run? In the legend there are only circle symbols, in the figure also squares. The way data are plotted suggests that Xact and ATONA measurements are bracketed. Can you first plot the Xact 100% data, followed by ATONA 100% data etc.?*

We have expanded the description of the measurement scheme in the caption and changed the figure as suggested. Thank you for catching the error in the legend.

*Figure 6. Maybe a matter of reader preference, but I prefer to see the 10 different analyses of one beam size plotted combined instead of interspersed. Now I find it difficult to see that variation within 10 similar experiments. And we are looking at ratios of blank corrected time zero intensities of 40 and 36? Inset is again rather small.*

The figures have been changed, and information requested added to the caption.

*Figure 7. The ARGUS RTIA data are from NMGRL? And are the measured with m/e36 on a Faraday with RTIA or multiplier? Colors of shaded lines are similar, not clear what they are showing.*

I added a reference to the paper section that describes the comparison datasets, which I think are too extensive to put in this already-lengthy caption. The shaded lines are an attempt to guide the eye to the many different groups of data points in the figure, and I have clarified this in the caption. I recognize that this figure is busy and that they are

hard to distinguish, but I think making them bolder would obscure the more-important data points themselves.

*Figure 8: in caption it is indicated that smaller aliquots are on the left, and larger on the right. Can you indicate different areas in the figure which are the 0.1cc, the 0.2cc aliquots etc. The NMGRL Argus measurements are with 40Ar on H2 with 10ˆ12 Ohm amplifier and 36 on L2 with 10ˆ13 Ohm amplifier? Do Argus data with 40Ar on H1 with 10ˆ12 or 10ˆ13 Ohm amplifier and 36Ar on L3 multiplier also exist? And if yes, how do they compare? Did NMGRL perform exactly the same experiment with 3 aliquots per pipette volume? And if not, what are the criteria to select these 3 data points?*

> The experimental protocol was (as close as possible) to identical during the APIS experiments, which hopefully will be published eventually. I have added this information to the caption. This represents the complete dataset. The requested clarification has been added to the figure.

*Table 1: what is the ± in the header row? 1SD? What is the 1-σ at the bottom of the table: the standard deviation of the ten measurements? Can you also report the mean (or the weighted mean)?*

> I have clarified the latter points and added a note in the caption stating that uncertainties are 1-sigma standard deviation.

*Figure A1: I don't like the interspersed way of plotting. Also with all the colors that look rather similar it is difficult to see what is what.*

> This has been changed.

*Figure A2: what is exactly plotted on the Y axis? Why not signal divided by average AX signal? Then the intercalibration factors mentioned in caption are immediately clear. And what is plotted on the X-axis? Why are there no data of aliquots 817, 820 etc. What is the beam size used for this intercalibration, is a baseline correction needed? And I'll assume data are regressed to time zero using a linear fit? What is the settling time, maybe worth mentioning, because a similar approach using RTIAs will take longer*

> This is exactly what is being done, only then the data are shown using delta units because the ratios are so close to unity. This is unclear, so I have changed the figure to simply show the ratios. The aliquot numbers represent extraction numbers from the air pipette, and the missing aliquot numbers were standards that were measured in typical multicollection mode in between intercalibration measurements. This is not meaningful outside of the lab and has been removed. The beam size is the full air standard described elsewhere; this has been clarified, along with details of the measurement scheme.

---

## Author Comment (AC2) · 6 May 2020

*Daniel Wielandt (Referee) wielandt@bio.ku.dk*

*General comments: This is a well-written article on an important novel amplifier technology called ATONA that provides an otherwise currently unavailable combination of low noise and high dynamic range for Faraday cup measurements of ion beams. The technology could significantly improve both current and future mass spectrometers, and is therefore of general interest to all mass spectrometry specialists. I however believe that its impact could be improved by including some additional information as mentioned below. Alternatively, the suggestions in general comments should be addressed in future publications.*

*The article focuses on comparing the ATONA to current 10E11, 10E12, 10E13 and a hypothetical 10E14 ohm amplifier, or rather their idealized Johnson-Nyquist noise characteristics, for the purpose of multicollector noble gas measurements. ATONA outperforms ideal i.e. model 10E13 ohm amplifiers with respect to signal-noise ratio for 10 second integrations which is (most likely) an appropriate integration time for many measurements, and approaches an ideal (and currently commercially unavailable) 10E14 ohm amplifier for a 100 second integration which is most likely to long to properly sample and back-project a noble gas ion beam evolution to T0. The high dynamic range and low noise-fast response is definitely an improvement as compared to traditional amplifiers. This versatility means that amplifiers do not need not be physically or electronically switched among Faraday cups for different applications, which is an additional advantage that complements their low-noise characteristics. An ATONA could also be useful for single detector instruments that still have merit due to the high sensitivities afforded by the small volumes of such instruments.*

*Although the comparison with traditional amplifiers at low signal intensities is appropriate, the paper could benefit from a more stringent comparison with ion counters where the noise characteristics at low signal intensities are dominated by Poisson i.e. counting noise of the individual ion arrivals. This noise is inherent to counting atoms or ions and cannot be avoided. An interesting question is therefore under which beam intensity x time i.e. accumulated charge conditions the "baseline" noise in an ATONA becomes comparable to this inherent and unavoidable counting noise that will also be present and superimposed on zero-beam i.e. electronic baseline noise? This would seems to be an appropriate lower dynamic range where ion counters would (decisively?) outperform ATONA in terms of precision (but not necessarily accuracy). This number could presumably be calculated based on the 1-10-100 second zero-beam measurements that have already been carried out. It may also be possible to tease out that information from e.g. figure 7, but it is better that it is presented.*

> It is true that the ATONA cannot replace ion counters for measuring very small signals. The baseline noise of the ATONA, while greatly reduced compared to other Faraday

amplifiers, is still larger than the near-counting-statistics noise level of an ion-counting multiplier. We show this, for example, in Figure 7, and we have added new discussion to the introduction clarifying the circumstances under which an ion-counting multiplier remains preferable. We also discuss in Section 3.2 that the ion counter remains necessary for very small signals on the NGX. We have also added a shot noise calculation to Figure 7.

*The paper would also benefit if the working principles of ATONA were more thoroughly discussed (without disclosing confidential information). The patent documents contain a lot of public information that could be condensed into a description of the technology. I think the mass spectrometry community would be more likely to adopt the technology if they could understand it better, rather than using it as a "black box" technology where one might run into an unpredictable problem. As a naive non-engineer I personally would like to know how leakage current is reduced. Is there a maximum charge than can be accumulated before "discharging" if that is even the appropriate term? Are there hysteresis effects in the capacitor that make it particularly hard to drive out or sense low buildups of charge that might adversely affect linearity at low signal intensities? Can charge buildup in the Faraday-amplifier system start to deflect incoming ions, changing the peak shape thereby affecting e.g. pseudo-resolving peak-shoulder measurements. Does the "firmware" make decisions on sampling rate or readout parameters, switching between different regimes that depend on beam intensities?*

We have endeavored to address these issues in the manuscript, and we address the specific points raised here in more detail in the response below.

Regarding leakage current: We assume that the reviewer is referring to the leakage current through the capacitor when there is non-zero charge accumulated. This leakage current is caused by migration of electric charges through the volume of the dielectric when an electric field is applied and creates non-linearity in measuring the accumulated charge over time, as part of the charge is lost through leakage during the measurement. Isotopx addressed this by first, the use of proprietary extremely low leakage dielectric for the feedback capacitor, then cooling the amplifier to reduce the already very small leakage current and then measuring its parameters to further compensate for the leakage. As a result, this error current is less than 1aA ($10^{-18}$A) for input currents above 1pA ($10^{-12}$A) creating <1ppm non-linearity. For smaller input currents the error current is reduces proportionally, $10^{-19}$A for 100fA ($10^{-13}$A), $10^{-20}$A for 10fA ($10^{-14}$A) and so forth, still maintaining <1ppm non-linearity. We have added this information to the manuscript.

Maximum charge: There is a maximum charge that can be accumulated by the feedback capacitor, which is determined by the value of the capacitor and the working voltage of the amplifier. However, the ATONA simply discharged the capacitor when it reaches the maximum value, a scenario that does not affect the measurement process. Only the rate of change of the transimpedance amplifier output voltage and therefore the rate of change of the accumulated charge is measured. This rate of change does not depend on the maximum charge value and the maximum value of the measured current depends only on the dynamic properties of the amplifier and subsequent data acquisition circuitry.

Hysteresis: Dielectric hysteresis may be defined as an effect in a dielectric material similar to the hysteresis found in a magnetic material. This causes a static shift in the

capacitor voltage for a certain charge dependent on the history of previous charges/discharges. Isotopx addressed this by the use of proprietary dielectric with paraelectric properties and with negligible hysteresis.  As a result, the effect of capacitance-voltage hysteresis on output voltage is unobservable.

Charge buildup: The Faraday buckets are directly connected to the input of the inverting amplifier. This fixes the voltage of the bucket at zero volts all the time regardless of the accumulated charge and therefore does not create any change or deflection in the incoming ions beams. We have added this information to the manuscript.

Firmware/black box decisions: No. The firmware neither changes any acquisition parameters, e.g. sampling rates, voltage ranges, measurement regimes or any other parameter, nor switches/changes any hardware values or components for any reason. This is done to preserve continuity, linearity, and repeatability of the measurements throughout the entire dynamic range.

*Throughout: The term Johnson-Nyquist noise is used in line 114, but then subsequent usage is about Johnson noise. Should abbreviate it JN-noise at first usage, and then refer to it as such subsequently.*

We have adjusted the text as suggested for better clarity.

*When discussing the performance using air and cocktail standards, it would be nice to have the approximate beam intensities tabulated in e.g. fA as that it the unit that is already reported for noise measurements.*

We have added this information where appropriate.

*Specific: First paragraph i.e. 8-22 could perhaps use a statement regarding engineering tradeoffs regarding multicollection versus volume/sensitivity, i.e. the increase in volume that tends to occur with multicollection and the related drop in sensitivity. This is one reason why single collector instruments still have a role. In fact, the versatility of the ATONA seems to make it very well suited for that role; this is only aided by its rapid response as discussed later.*

We agree with both the principle of the statement regarding the value of small volume instruments and with the possible role of the ATONA in such instruments, both because of its dynamic range and because of its response time. We have added statements to this effect to the second paragraph of the manuscript.

*Second paragraph, line 30. Mention of long settling time for high value resistors is relevant in case of dynamic measurements, but static multi-collection of noble gases all but removes the settling time issue since any single resistor only measures one very slowly evolving beam. This should be mentioned in order to be fair to the current generation of high-ohm multi-collector equipped instruments.*

While it is true that settling time is less important for multicollector instruments, it can be long enough on some high-gain RTIAs that without mitigation it affects the settling time of the measurement on the time scale of gas inlet. We have added this caveat to the manuscript.

*Paragraph 6, Line 80. Could the authors perhaps make a back of the envelope error propagation calculation of how much of the air correction error on a blank subtraction on their instrument would arise from the 36Ar using a ion counter versus the ATONA? Or conversely the calculations suggested in the general comments regarding comparison of counting noise vs zero beam noise? This would be highly relevant for e.g. Ar or Ne dating of young samples where samples or fractions may be comparable in intensity to blanks.*

We have added such a calculation for a typical young basalt sample.

*Paragraph 7, Line 84. If possible, it would be nice if the patent were hyperlinked.*

We have provided a hyperlink.

*Paragraph 9 A formulation of Johnson-Nyquist noise with some appropriate reference and description would be useful for non-specialists.*

We have added additional information.

*Paragraph 11, Lines 130-140. This is a bit hard to read, and the reporting would benefit from a data-table showing the noise characteristics for 1, 10 and 100 second integrations with ATONA and 10E11-14 resistors. In such a way, one could focus on describing the noise "crossover" points for the various detector technologies that most readers would be searching for anyway as seen in figure 4.*

We have added the requested table as Table 1.

*Figure 6 (and figure A1) It is hard to identify the ranges, could the color code somehow be complemented by a change in marker style? It might also be a good idea to write the ranges as from 200% to 0.36% rather than between 200% and 0.36%.*

In response to this comment and a comment from Kuiper, we have reorganized the figures so that the analyses are grouped by signal size rather than by analysis order, which we hope will also address the difficulty in distinguishing them from one another. The point about the ranges being inclusive is noted and this change has also been made.

*Figure 7 We should expect a number of inflection points where all faraday mass spectrometer technologies gradually switch to follow a slope determined by counting noise (N^0.5) rather than signal over "baseline" JN or kTC noise (N^1). The linear error envelopes could give the erroneous impression that Faraday-based technologies can eventually outperform counting noise at high intensities, this should be avoided.*

We have added a line showing the calculated uncertainty limit for a time-zero regression through data affected only by shot noise.

*Table 1 The table should include the intensity of the smallest ion beam intensity i.e. the 36Ar intensity in fA. It would also be nice to have (and discuss) an MSWD to compare internal*

*precision and external reproducibility for all the measurements. A calculated average for the different intensities would also be nice, and could be plotted to evaluate non-linearity.*

> The uncertainties shown at the bottom of the table are in fact the population uncertainties. We have added a line also showing the averages as requested, which should also make it more clear that what is meant by the 1-sigma uncertainties at the bottom (this is also in response to a comment by Kuiper). The averages are all well within uncertainty of one another, but an experiment aimed at properly assessing isotope ratio linearity would require many more measurements for the smaller signal sizes.

*Table A1 The table should include the intensity of the smallest ion beam intensity i.e. the 38Ar intensity in fA. It would also be nice to have (and discuss) an MSWD to compare internal precision and external reproducibility for all the measurements. A calculated average for the different intensities would also be nice, and could be plotted to evaluate non-linearity.*

*Data for 0.36% measurements seem improbably precise, are they missing a digit*

> See notes from above. And yes, thank you for catching that—the table was hanging off the edge of the page, truncating the internal uncertainties for these measurements. The tables have been modified to fit the page better.

*It would also be nice to discuss the presumably significant decrease in precision when going from 5.2% aliquots to 2.6% aliquots and lower. Is this a characteristic of ATONA, or is it due to error propagation effects from subtraction of blank 38Ar + H37Cl?*

> While the uncertainties increase for smaller signal sizes, the relationship is as expected and is primarily governed by the measurement uncertainty of the smaller ion beam. We discuss this further in Section 3.2 and show it in Figure 7.

---

## Author Comment (AC3) · 6 May 2020

We greatly appreciate the thoughtful comments from referee Trevor Ireland, an expert in geochemistry and mass spectrometry who has worked with capacitor-based ion detection systems on the SHRIMP ion microprobe. The review is reprinted below in its entirety in italics with author responses shown by indentation where appropriate.

*Referees Comment on ATONA Charge mode system.*
*Trevor Ireland, RSES ANU.*

*This is a timely paper concerning the introduction of the ATONA charge mode collection system to an ISOTOPX NGX.*

*The challenge of getting the the capacitative feedback system to work has been ongoing for a number of years. Both Cary and Keithley electrometers have had the ability to measure charge accumulation, but getting it to work in a routine analytical setup has been somewhat difficult (see Esat 1995, and Ireland et al. 2014).*

> We have added a discussion of this background work to our introduction.

*The physics behind charge mode and the implications for data collection are interesting, but are not particularly well explained in this paper. Potentially this is because of the patent that is being sought for this system. In any case, the issues concerning the noise floor for a capacitive system effectively relate to a "read" noise in the capacitor system as opposed to the Johnson Noise in a resistive system. Hence, there is a constant noise component in the capacitative read system (see Ireland et al. 2014), and the longer you integrate the better the signal to noise. On the other hand, you continually integrate Johnson Noise and so the signal/noise does not improve as quickly for an increase in integration time. For comparison, we have set up our capacitative system for a 2 s integration and the noise is similar to the 10e-13 ohm resistor, pretty much similar to what is achieved here. The 2s integration is appropriate for an ion microprobe because of the continual change in analytical conditions. A longer integration time is fine for a noble gas instrument because the gas is effectively homogenised in the source and there are only longer term fractionation processes to deal with.*

> We have attempted to further clarify the operation of the ATONA system in light of these comments, but within the restrictions imposed by Isotopx' trade secrets. The constant noise component highlighted here is one of the advantages of the ATONA compared to the conventional resistor amplifier. In ATONA signal-to-noise ratio increases much faster with an increase in the integration time compared to the resistive amplifier. As suggested here, even longer integration times will be beneficial for more stable systems such as TIMS and stable isotope gas source mass spectrometers.

*The work in the Isotope NGX is based around noble gas analysis, and specifically Ar isotopes. On one hand, this is a good system to look at because there is a good dynamic range in the isotope ratios under consideration. It also has the benefit that the ion beam is only changing at a (slow) steady rate allowing a good description of the progression of the counting statistics. The data show that the system performs well at the level commensurate with the measured ratios. On the other hand, Ar isotopes are not typically measured to high precision (e.g. as might be achieved for TIMS or ICP-MS analysis, or even SIMS analysis). This makes it also more difficult to establish the linearity of the system as well.*

This is correct, although the measurements of stable zero-beam noise levels we present are an indication of the performance limits of the ATONA in a mass spectrometer with a more stable ion beam such as a TIMS. We expect results from TIMS labs with ATONA amplifiers to be published soon.

*It is evident that the noise floor is still an issue for the 36Ar measurements described here. So as the volume of Ar gas is reduced, the error magnification from measuring the 36Ar/40Ar and resulting corrections to 40Ar/39Ar are still going to be a limitation and will likely still need to be carried out on an electron multiplier.*

We do agree that the noise floor of the ATONA is comparable to an analog electron multiplier, but is not competitive with an ion counting electron multiplier for very small signals.

*The benefit of the capacitative system is that measurements can be carried out on more Faraday cups, and potentially without the need for an electron multiplier (ion counter). The noise floor we have achieved is better than 500 c/s which means that for most isotope ratio measurements Charge mode is adequate and very often superior to an electron multiplier. At the upper level of count rates, 250-1000 x 10e3 c/s, the charge mode seamlessly connects with 10e12 ohm resistor capability. At the lower end, we have measured isotope ratios down to 10,000 c/s, which is well removed from the gain drift and dead time issues of an electron multiplier. As such charge mode does provide that connection between electron multipliers and the traditional resistor feedback amplifiers. But as demonstrated in this paper, it is a complementary aspect of the measurement of isotope ratios in geochemistry.*

*References*

*T.M. Esat, Charge collection thermal ionization mass spectrometry of thorium. International Journal of Mass Spectrometry and Ion Processes 148 (1995) 159–170.*

*T.R. Ireland, N. Schram, P. Holden, P. Lanc, J. Ávila, R. Armstrong, Y. Amelin, A. Latimore, D. Corrigan, S. Clement, J.J. Foster, W. Compston, Charge-mode electrometer measurements of Sisotopic compositions on SHRIMP-SI. International Journal of Mass Spectrometry 359 (2014) 26– 37.*